# The Prevalence of HPV in Oral Cavity Squamous Cell Carcinoma

**DOI:** 10.3390/v15020451

**Published:** 2023-02-06

**Authors:** Seyed Keybud Katirachi, Mathias Peter Grønlund, Kathrine Kronberg Jakobsen, Christian Grønhøj, Christian von Buchwald

**Affiliations:** Department of Otorhinolaryngologi, Head & Neck Surgery and Audiology, Rigshospitalet, 2100 Copenhagen, Denmark

**Keywords:** oral cavity squamous cell carcinoma, oral cavity cancer, head and neck cancer, human papillomavirus, HPV genotype, prevalence, global, worldwide, systematic review, meta-analysis

## Abstract

Human papillomavirus (HPV) is an important risk factor in a subset of head and neck squamous cell carcinomas (HNSCC), but the association with oral cavity squamous cell carcinomas (OCSCC) remains controversial. This study aimed to identify the prevalence of HPV infection in OCSCC. A systematic search on PubMed and EMBASE was performed, including articles assessing the prevalence of HPV-positive (HPV+) OCSCC published from January 2017 to December 2022. OCSCC was considered HPV+ by the detection of HPV DNA, HPV RNA, and/or p16 overexpression in the tumor mass. A meta-analysis was made determining the overall HPV+ OCSCC prevalence. We included 31 studies comprising 5007 patients from 24 countries. The study size ranged from 17 to 940 patients. The HPV+ OCSCC proportion variated widely and ranged from 0% to 37%. Tumors in the tongue were the predominant sublocation for HPV in the oral cavity. The meta-analysis revealed that the overall HPV+ OCSCC prevalence is 6% (95% CI; 3–10%), and only one study found HPV and OCSCC significantly associated. Thus, HPV may not be a necessary or a strong risk factor in OCSCC oncogenesis, and the possibility of a site misclassification of a mobile tongue with the root of the tongue cannot be excluded.

## 1. Introduction

Oral cavity cancer is the most frequent cancer subsite in the head and neck area with more than 377,700 cases worldwide in 2020, placing them as the 16th most common cancer overall, the 11th most prevalent cancer in men, and the 18th most frequent cancer in women [1]. In addition, human papillomavirus (HPV) is widely prevalent worldwide and is the most frequent sexually transmitted infection in the United States [2,3]. It is well established that HPV is an important risk factor in a subset of head and neck squamous cell carcinoma, particularly oropharyngeal squamous cell carcinomas (OPSCC) [4,5,6,7].

Although well-known risk factors of oral cavity cancer are the consumption of tobacco (both smoked and smokeless), alcohol, and poor oral hygiene [8], the association of HPV as a risk factor for oral cavity squamous cell carcinomas (OCSCC) remains controversial since the association was described for the first time in 1983 [9].

HPV can be classified into either a low-risk HPV or high-risk (HR) HPV regarding oncogenicity. HR-HPV genotypes currently known in oropharyngeal tumors are HPV16, 18, 26, 31, 33, 35, 45, 56, 58, 59, and 67 [10]. The predominant HR-HPV genotype in oropharyngeal tumors is HPV16, accounting for 86% of HPV+ OPSCC [10].

There are no universal HPV-testing methods available for OCSCC. The identification of HPV E6/E7 mRNA expression is considered the golden standard for measuring HPV infection by some authors, as this technique detects oncogenic transcriptionally active HPV. However, the method is expensive, technically challenging to perform, and requires fresh frozen tumor material, which is not always collected [11,12,13]. The World Health Organization (WHO) has stated that the diffuse cytoplasmic and nuclear immunoreactivity for p16 may act as a trustworthy surrogate marker for the incidence of HR-HPV in OPSCC, but the method has only been validated for OPSCC and does not apply to OCSCC [14]. Additionally, p16 does not seem to have the same role in OCSCC as in OPSCC. The overexpression of p16 has demonstrated poor performance as a prognostic marker for overall survival in OCSCC, and it has been unrecommended to use as a tool for OCSCC in study trials [15]. It has correspondingly been suggested to improve HPV detection accuracy by combining p16 immunohistochemistry and HPV DNA polymerase chain reaction analysis since this combination increases sensitivity and specificity [13,16,17].

This systematic review seeks to identify the burden of HPV infection in OCSCC globally by examining the prevalence reported in the most recently published papers.

## 2. Materials and Methods

### 2.1. Systematic Literature Search Strategy

In December 2022, a systematic search was last updated by one author (S.K.K) using the databases PubMed and EMBASE for articles regarding the prevalence of HPV+ OCSCC published between January 2017 and December 2022.

Studies including 10 or fewer patients were excluded alongside other systematic reviews and meta-analyses. Furthermore, we excluded studies if: (1) it was unclear whether the study addressed the oral cavity alone or oropharynx as well or (2) it concerned the HPV+ OCSCC prevalence in specific subpopulations stratified by gender, comorbidities, or ethnicity. An OCSCC was considered HPV+ if HPV DNA, HPV RNA, and/or p16 overexpression were detected in the tumor mass. If there was more than one way to define HPV positivity in a study, we subtracted the result in the following order: (1) double/triple positivity, (2) HPV RNA, (3) HPV DNA, and (4) p16 overexpression.

This systematic review followed the procedures of the Preferred Reporting Items for Systematic Reviews and Meta-Analyses (PRISMA) statement [18]. The title and abstract of the studies were screened for eligibility while a second screening assessed the full-text. In the case of doubt regarding whether a study met the inclusion criteria, it was discussed with the author group.

The following search strategy was used when searching PubMed: (“Head and neck” or “oral cavity” or oral or mouth or lips or “buccal mucosa or tongue or “hard palate” or gingiva) AND (squamous cell carcinoma* or squamous cell neoplasm*) AND (HPV or “human papillomavirus” or “human papilloma virus” or p16) AND (prevalence or frequency or incidence). An asterisk at the end of the word means that all possible suffixes of a word are included in the search. MeSH terms (medical subject headings) were included as well: “carcinoma, squamous cell”, “neoplasm, squamous cell”, human papillomavirus, papillomaviridae. The search was restricted to the English and Danish languages.

The same keywords and Emtree (medical subject headings) were used to create four different searches in Embase, which were combined with “AND”. The search was limited to English language and articles published between 2017–2022:“Head and neck” or oral or “oral cavity” or mouth or lips or “buccal mucosa” or “hard palate” or gingiva or tongue;squamous cell carcinoma/or Squamous cell carcinoma* or squamous cell neoplasm*;Wart virus/or HPV or “human papillomavirus” or “human papilloma virus” or p16;Prevalence or incidence or frequency.

The following data were extracted from the included studies: publication year, country, female to male ratio, HPV status, anatomical sublocation, definition of HPV-positivity, mean age, and sex.

### 2.2. Statistical Analysis

A meta-analysis was made to determine the overall HPV+ OCSCC prevalence of the included studies. The prevalence was expressed as relative risk in the random-effects model, which was conducted due to the wide variation in the prevalence. The meta-analysis and forest plot were conducted using the software R (version 4.2.2) and the packages “meta”, “metafor”, and “forestplot”.

## 3. Results

The search in PubMed and Embase databases retrieved 1303 studies after duplicates were removed, shown in Figure 1. Thirty-one studies met the inclusion criteria, comprising 5007 patients with OCSCC from 24 different countries. Sixteen studies were from Asia [19,20,21,22,23,24,25,26,27,28,29,30,31,32,33,34], eight from Europe [35,36,37,38,39,40,41,42], two from Africa [43,44], two from South America [35,45], two from North America [46,47], and one from Asian Pacific [48]. The study size ranged from 17 patients in a study from Uganda [44] to 940 patients in the Netherlands [41]. There was no congruity in the size of the population samples regarding geographical areas.

### 3.1. Clinical Characteristics of HPV+ OCSCC

Five studies (*n* = 112) reported the mean age of patients with HPV+ OCSCC varying from 53 to 63 years. Two studies reported that HPV+ OCSCC was significantly associated with a lower mean age than HPV-negative (HPV−) OCSCC (*n* = 59) [38,41] while five studies found it statistically insignificant [24,27,28,30,45]. The mean age in HPV−OCSCC ranged from 57.5–64 years [37,38,41,45,49]. The overall mean age was reported in thirteen studies ranging from 45–70 years, shown in Table 1.

The female-to-male ratio varied from 1:1 in the Republic of Korea to 1:8 in Réunion Island among patients with OCSCC [23,48]. Three studies reported a significant association between patients with HPV+ OCSCCs and the male gender [37,41,49], whereas no association was reported in five studies [24,27,28,30,45].

One study (*n* = 26) found HPV+ OCSCC to be associated with less tobacco smoking and/or chewing [37] while six studies did not find an association [24,27,28,30,38,45].

A higher alcohol consumption and a higher number of sexual partners were demonstrated to be associated with HPV+ OCSCC in one study [38]. Two studies found an HPV association with an earlier T-stage [41,45].

Data regarding HPV+ OCSCC association with clinical characteristics could not be extracted from seventeen studies either because it was not described or the data was pooled on multiple anatomical locations in HNSCC [19,20,22,24,25,26,29,31,32,33,39,40,42,43,44,46,47].

### 3.2. The HPV+ OCSCC Prevalence Worldwide

The lowest prevalence of HPV+ OCSCC was 0% found in the Philippines [20], the United Kingdom (UK) [36], India [19,22], the Republic of Korea [25], and France [40]. The highest prevalence was reported to be 37% in Jordan [24]. Additionally, both low and high HPV occurrences were reported across all geographical locations. A study from India was the only study that concluded a statistically significant association of OCSCC to HPV infection by a Chi-Square Test [28], shown in Table 1.

A proportional meta-analysis was conducted, determining the total prevalence of HPV+ OCSCC to 6% (95% CI; 3–10%). There was a great heterogeneity in the prevalence as well, I^2^ > 75%, *p* < 0.01, shown in Figure 2.

The HPV status in the tumors located in different anatomical subsites of the oral cavity was specified in 16 studies (*n* = 284). Eight studies observed the highest proportion of HPV+ OCSCC in the tongue (*n* = 162) [19,24,27,28,30,35,42,45], and in three studies, the highest prevalence was seen in the floor of mouth (FoM) (*n* = 89) [37,38,41]. The HPV+ OCSCC proportion in the tongue varied from 0% to 100%, and among patients with tumors in the FoM, it varied from 0% to 57.1%. The specificity of the mobile tongue has been indicated as follows: * specified mobile tongue, ** specification questionable due to not following a classification system or only writing “tongue”, *** unspecified due to inclusion of overlapping lesion of tongue and/or lingual tonsil and/or tongue unspecified, shown in Table 1.

### 3.3. HPV Detection Methods and Definitions

Multiple different HPV detection methods were used in the included studies. Fifteen of the studies were based on double/triple positivity (*n* = 3743) [21,25,27,28,29,31,32,34,37,39,40,42,46,47,50], eleven based on HPV DNA (*n* = 1001) [20,22,24,26,30,33,36,38,43,45,49], two on HPV RNA (*n* = 140) [19,35], and three on p16 IHC alone (*n* = 123) [23,44,48], shown in Table 1. The definition of p16 IHC positivity varied among the studies. Most of the studies defined p16 overexpression as positive if staining ≥70% tumor cells were observed according to the guidelines from the College of American Pathologists [51], shown in Table 1. Both high and low HPV+ OCSCC proportions was demonstrated regardless of the detection methodology, shown in Table 1.

### 3.4. HPV Genotypes

Eighteen studies reported the specific HPV genotypes (*n* = 130) [19,21,24,25,26,27,28,29,32,34,37,39,41,42,43,45,47,49]. HPV16 was the predominant subtype in fourteen of the studies (*n* = 171) [21,24,25,26,28,29,32,39,41,42,43,45,47,49] and was observed in up to 100% of the HPV+ OCSCC cases in three studies (*n* = 17) [24,39,45]. Another common genotype was HPV18, which was the most frequent subtype in three studies (*n* = 28) [27,30,34] and was observed in up to 100% HPV+ OCSCC in one study (*n* = 8) [34]. The pooled prevalence of other HR genotypes (HPV31,33,45,52,59) presented in the enrolled studies was 6.92% (*n* = 9). Co-infections were reported in four studies of which HPV16 was present in most cases and was coinfected with HPV18 (*n* = 25), HPV31 (*n* = 1), and HPV39 (*n* = 1) [25,27,28,30].

## 4. Discussion

We conducted a systematic review and meta-analysis that examined the HPV+ OCSCC prevalence worldwide. Thirty-one studies were included with a total of 5007 patients. There was a wide variety in the prevalence of HPV+ OCSCC reported, with the lowest prevalence being 0% in the Philippines, the UK, India, the Republic of Korea, and France and the highest being 37% in Jordan [20,22,24,25,33,36,38,40]. Studies with a higher HPV+ proportion had greater statistical uncertainty as demonstrated in our meta-analysis, shown in Figure 2. The Jordanian paper consisted of a small study size comprising 27 patients, and they only detected HPV DNA, which should be noted. Furthermore, they found no HPV association with OCSCC or other clinical factors like tobacco, alcohol, age, or gender [24]. The studies that reported 0% HPV+ OCSCC proportion had a sample size ranging from 31 to 166 individuals and defined HPV+ as double positivity with p16+/HPV DNA or solely HPV DNA. The study from India observed the HPV+ OCSCC fraction to be 13% and was the only study to find a statistically significant association between the HPV status and OCSCC [28]. Moreover, with the HPV−OCSCC being more frequent than HPV+ OCSCC in all the studies and with our meta-analysis determining the total prevalence of HPV+ OCSCC globally being 6% (95% CI; 3–10%), it may indicate that HPV infection is not a mandatory nor a strong risk factor and does not constitute a high proportion of the OCSCC worldwide. Additionally, a study found no difference in survival outcomes between patients with HPV+ OCSCC and HPV−OCSCC when stratified on p16 overexpression status [15]. Regarding survival, our enrolled studies support that neither p16-status nor HPV-status have an impact on OCSCC patients [32,35,37,41]. Only one study reported a trend towards increased survival in HPV-positive individuals, although they did not find it statistically significant [42].

HPV+ OPSCC is more prevalent in the Western world [52,53], but in our enrolled studies there was no correlation, both high and low HPV+ OCSCC prevalence were observed regardless of the geographical area. Additionally, studies have shown HPV+ OPSCC to be more apparent among the younger patients that generally consuming less alcohol or tobacco [54,55]. Compared to HPV+ OCSCC in our study, only two enrolled studies reported such significant association with younger individuals [38,41]. Solely, one study observed HPV+ OCSCC association with less tobacco consumption [37]. A higher alcohol consumption and number of sexual partners were demonstrated to be associated with HPV+ OCSCC in one study as well [38]. Interestingly, the same study demonstrated a significant HPV+ OCSCC association to an earlier T-stage and concluded that it may indicate that both alcohol intake and oral HPV infection act synergistically, explaining earlier tumor onset. Overall, this could emphasize that the HPV+ OCSCC is most frequently seen in elderly patients, and the association with less alcohol and tobacco consumption is weak in contrast to the HPV+ OPSCC. Presumably, the oncogenicity of HPV-infection is of less magnitude in the genesis of OCSCC compared to OPSCC. This is supported by the decrease in incidences of OCSCC (i.e., in the USA while HPV-related OPSCC incidents have been increasing) [56].

We found that HPV16 was the predominant genotype and HPV18 was the next common genotype in the oral cavity. Other HPV HR-subtypes (HPV 31,33,45,52 and 59) were less frequent, but as a shortcoming, some studies only examined the presents of HPV16 and/or HPV 18. The 9-valent HPV vaccine (Gardasil9) may in theory reduce the possible HPV-caused carcinogenesis in the oral cavity since it covers all these subtypes except for HPV 59, which was only present in one case [49].

The anatomical subsites of HPV+ OCSCC was specified in 14 of the studies. Discrepancies were observed regarding the most prevalent sublocation. Ten studies did not investigate the subsites of the tumors, which complicates the interpretation of the results and is a limitation for determining the true representation of HPV+ OCSCC sublocations [23,29,31,37,39,43,44,46,47,48]. However, most of the included studies reported that HPV+ OCSCC were more apparent in the tongue followed by the FoM. This raises an interesting issue as to whether some of the tumors in the tongue might have arisen from the base of the tongue, which is classified as a part of the oropharynx instead of the mobile tongue, which is considered a subsite of the oral cavity. 

Hence, a shortcoming can be attributed to the different definitions of the oral cavity among the studies regarding the included subsites. The Surveillance, Epidemiology, and End Results (SEER) database and Systematized Nomenclature of Medicine—Clinical Terms (SNOMED-CT) do not distinguish between the base of the tongue and the mobile tongue when describing the oral sites but labels them both as lingual or tongue, which can cause some anatomical site misclassification and contribute to the confounding of the reported HPV+ OCSCC proportion [57,58]. Furthermore, SNOMED-CT consider the palate as a part of the oral cavity without distinguishing between the hard palate and the soft palate, with the latter being a part of the oropharynx [58]. Our study demonstrates some outlying studies that report a higher prevalence of HPV+ OCSCC than the pooled prevalence, and a high proportion of the tumors are found in the tongue [19,24,28,30,38]. In the paper with the highest reported HPV+ OCSCC proportion, we observed that all HPV+ cases were seen in the tongue, and 50% (10/20) of all the tongue tumors were HPV+, shown in Table 1. Nonetheless, they distinguished between the base of tongue and the mobile part, but considering the ICD-10 codes, overlapping lesions might have been involved (C.02.8 “overlapping lesion of tongue”) as they do not report the exact included ICD-10 codes [59]. Another example to notice is that the only study that found a significant association between HPV and OCSCC included the soft palate as a part of the oral cavity, as shown in Table 1. Moreover, we excluded some relevant studies due to the unclear definition of the term “oral”, which might solely refer to the oral cavity or the combination of the oral cavity and oropharynx. On the contrary, these mistakes should easily be avoided as the anatomical sites are clearly defined [59,60]. A previous study has investigated the prevalence of HPV in palatine tonsillar squamous cell carcinoma, subdivided, according to the certainty of tonsillar tumor origin into specified tonsillar squamous cell carcinoma (STSCC) and non-specified tonsillar squamous cell carcinoma (NSTSCC). The study observed the proportion of HPV+/p16+ to be 72% for STSCC and 21% for NSTSCC [61]. Hence, it would be interesting to explore if there was a big difference in the prevalence of HPV+ tumor according to the certainty of mobile tongue origin as well. Eventually, the HPV+ OCSCC fraction globally might be smaller than the 6% determined in this study.

The inconsistency of HPV detection methods among the studies was remarkable and a limitation to this review. PCR, ISH, hybrid capture 2 high-risk HPV (HC2-HPV DNA), and p16 IHC were among the different methods applied in the studies. We prioritized results from the methods that have defined HPV presence by double positivity since studies based on HPV DNA and p16 combined are more reliable, as the method has demonstrated a high sensitivity and specificity when considering HPV16 E6/E7 mRNA detection as the golden standard [13,16,17]. Many of the enrolled studies only detected HPV DNA with PCR, which is not the equivalent to the virus being transcriptionally active and could lead to false-positive test results, making this method suboptimal [17,62]. A few studies were based on p16 IHC positivity, and they all reported a higher HPV+ OCSCC prevalence than the pooled result determined by our meta-analysis. The overexpression of p16 might not reveal the true result, as it is only identified as valid for OPSCC, where it furthermore yields false-positive results in up to 20% when compared to the detection of HPV DNA [50,62]. It has also been recommended not to use p16 IHC as a tool for OCSCC in study trials since it has shown poor performance as a prognostic marker for overall survival in OCSCC [15].

To compare the studies and determine the HPV+ OCSCC fraction worldwide, it is important to find a homogenous HPV detection method that is both highly sensitive and specific for biologically active HPV and considered beneficial in the cost-effective context.

Moreover, there is no consistency in the definition of p16-positivity. Most of the studies defined p16 IHC overexpression as positive if staining ≥70% of the tumor cells was observed. Other studies set the limit of the staining for p16 to be positive at >75%, >50%, and >10%, and one study used a 5-tiered point system to determine p16+. The wide variety of p16-positive definition can also result in a wrong interpretation of the HPV+ OCSCC proportion.

Lastly, a meta-analysis was conducted, revealing significant heterogeneity I^2^ > 75%, *p* < 0.01, and the global burden of HPV+ OCSCC to 6% (95% CI; 3–10%). (Figure 2). We did not stratify for the various detection methods because some of the detection methods were too few for a valid meta-analysis, which is a limitation that should be noted. We excluded from our review studies and meta-analyses that assessed the prevalence of HPV in the head and neck area if data from the oral cavity could not be extracted due to pooled data from multiple anatomical locations. Hence, it entails some missing data that could be used in our study.

To the best of our knowledge, with over 5000 patients and 31 studies, this is the largest systematic review and meta-analysis conducted evaluating the association of HPV infection in solely OCSCC worldwide within the last five years.

## 5. Conclusions

In conclusion, there was a significant heterogeneity in the HPV+ OCSCC prevalence worldwide, varying from 0% to 37%. Studies with higher HPV+ fraction had greater statistical uncertainty. The most prevalent HPV genotype was HPV 16, and the second most prevalent was HPV 18. Tumors in the tongue were the predominant sublocation for HPV in the oral cavity. The meta-analysis revealed the pooled HPV+ OCSCC prevalence worldwide to be 6% (95% CI; 3–10%) from the included studies. HPV−OCSCC were more frequent than HPV+ OCSCC in all the studies and only one study found HPV significantly associated with OCSCC. Thus, taking the heterogeneity and limitations into account, HPV may not be a necessary factor nor a high-impact risk factor regarding OCSCC development, and the possibility of site misclassification of the mobile tongue with the root of the tongue cannot be excluded.

## Figures and Tables

**Figure 1 viruses-15-00451-f001:**
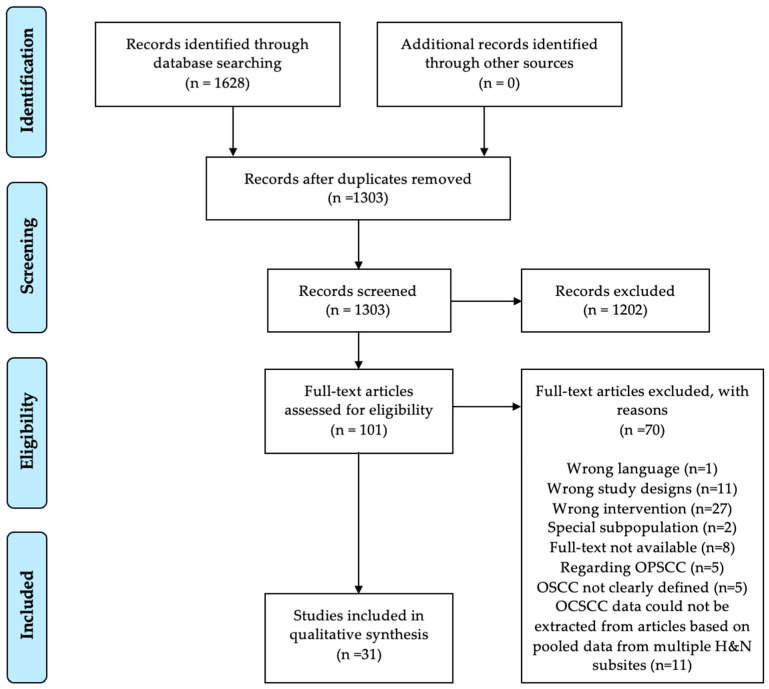
Prisma Flowchart. OCSCC: oral cavity squamous cell carcinoma, OPSCC: oropharyngeal squamous cell carcinoma, OSCC: oral squamous cell carcinoma, H&N: Head and Neck.

**Figure 2 viruses-15-00451-f002:**
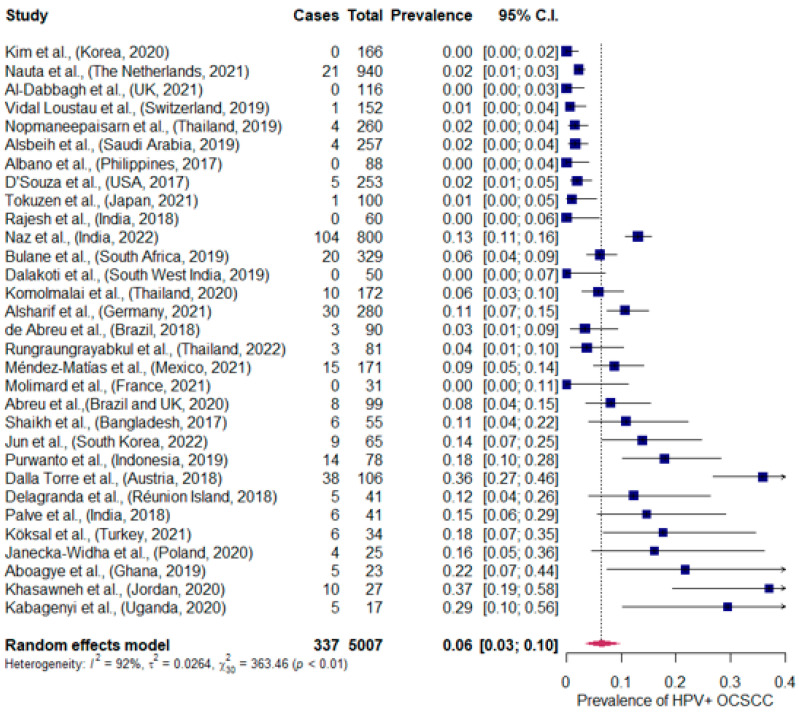
Meta-analysis of the HPV+ OCSCC prevalence in the included studies, representing the present HPV+ OCSCC prevalence worldwide [19,20,21,22,23,24,25,26,27,28,29,30,31,32,33,34,35,36,37,38,39,40,41,42,43,44,45,46,47,48,49]. HPV+: human papillomavirus positive. OCSCC: oral cavity squamous cell carcinoma. 95% C.I: confidence interval.

**Table 1 viruses-15-00451-t001:** HPV-positivity in OCSCC among patients worldwide.

Authors (Area and Publication Year)	Centre/Data Source	Study Period	Total Cases		Mean Age		F:M Ratio in Total	Share of HPV+ Patients in %	OCSCC Sublocations (Share of HPV+ Patients in %)	Definition of HPV Positivity Based on
				HPV+	HPV−	Overall				
**Africa**										
Ghana, 2019 [43]	Tertiary Hospital, Kumasi	2007–2016	23	-	-	**-**	-	22	-	HPV DNA PCR
Réunion Island, 2018 [48]	Commission Nationale d’Informatique et Liberté	2009–2013	41	-	-	-	1:8.3	12.2	-	p16 IHC (>10%)
South Africa, 2019 [49]	Universitas Academic Laboratories in Bloemfontein	2004–2014	329	57.2	58.5	-	1:3.1	6.1	-	HPV DNA PCR
Uganda, 2020 [44]	Uganda Cancer Institute	2018–2019	17	-	-	-	-	29.4	-	p16 IHC (>70%)
**Asian-Pacific**										
Bangladesh, 2017 [31]	Dhaka Medical College Hospital, A.I. Khan Laboratory and Millennium Dental Clinic in the city	2014–2016	55	**-**	**-**	**-**	**-**	10.9	-	HPV DNA PCR and p16 (>50%)
India, 2018 [19]	Institute of Bioinformatics and Applied Biotechnology	-	41	-	-	-	1:2.9	15	Oral tongue (83.3%) **, buccal mucosa (16.7%)	HPV RNA PCR
India, 2018 [33]	R. L. Jalappa Hospital and Research Centre, Kolar, Karnataka	-	60	-	-	53.7	3:1	0	0%	HPV DNA PCR
India, 2022 [28]	A tertiary care hospital	-	800	-	-	45.2	1:7.1	13 ^a^	2/3rd of tongue (32.7%) **, FoM (1.9%), lips (1.9%), buccal mucosa (31.7%), soft palate (9.6%) ^a^, hard palate (5.8%), retromolar trigone (4.8%), gingivobuccal sulcus (6.7%), alveolus (4.8%)	HPV DNA PCR and p16 IHC (>75%)
Indonesia, 2019 [30]	The Dharmais National Cancer Hospital, Jakarta	2003–2013	78	-	-	47.1	1:1.5	17.9	Tongue (74.4%) **, buccal (2.6%), gingiva (3.8%) maxilla (2.6%), mandible (3.8%), palate (5.1%), lips (7.7%)	HPV DNA PCR
Japan, 2021 [32]	Ehime University Hospital	2004–2013	100	-	-	70.3	-	1 ^c^	Tongue (10%) **, maxillary gingiva (20%), mandibular gingiva (30%), FoM (20%), buccal mucosa (10%), lip (10%) ^d^	HPV RNA PCR and p16 IHC (>70%)
Korea, 2020 [25]	Catholic Medical Center hospitals, Seoul St. Mary’s Hospital and Bucheon St. Mary’s Hospital	2011–2019	166	-	-	-	-	0	0%	HPV DNA PCR and p16 IHC (>70%)
Philippines, 2017 [20]	Mariano Marcos Memorial Hospital and Medical Center	2003–2013	88		-	-	-	0	-	HPV DNA PCR
Republic of Korea, 2022 [23]	Tertiary University in Seoul	2008–2020	65	-	-	60	1:1	13.8	-	p16 IHC (>75%)
South West India, 2019 [22]	Manipal Academy of Higher Education	2015–2017	50	-	-	53.7	1:3.2	0	0%	HPV DNA PCR
Thailand, 2019 [29]	King Chulalongkorn Memorial Hospital Bangkok	2010–2016	260	-	-	61.3	1:1.4	1.5	-	HPV DNA ISH and p16 IHC (>70%)
Thailand, 2020 [27]	Chiang Mai University and Prince of Songkla University and Chulalongkorn University and Khon Kaen University	1999–2019	172	-	-	66	1.2:1	5.8	Tongue (50%) **, FoM (10%), gum/alveolar mucosa (30%), buccal/labial mucosa (0%), retromolar area (10%), other sites (0%)	HPV DNA PCR and p16 IHC (5-tiered point system)
Thailand, 2022 [34]	Mahidol University, Bangkok. Faculty of Dentistry, Khon Kaen University.	2013–2019	81	-	-	-	1.5:1	3.70	Mobile tongue (29.6%) **, FoM (4.9%), gingiva (32.1%), buccal mucosa (13.6%), hard palate (7.4%), retromolar trigone (6.2%), lip (6.2%) ^b^	HPV DNA PCR and p16 IHC (>70%)
**Europe**										
Austria, 2018 [38]	Medical University of Innsbruck	2008–2012	106	53.32	60.76	58.9	1:2	35.8	Tongue (23.7%) **, FoM (31.6%), alveolar mucosa (15.8%), cheek (13.2%), palate (15.8%)	HC2-HPV-DNA
France, 2021 [40]	Besançon University Hospital	2005–2018	31	-	-	-	-	0	0%	HPV DNA PCR and p16 IHC (>70%)
Germany, 2021 [37]	The University Medical Center of Lübeck	2002–2011	280	63.3	62.8	62.8	1:2	10.7	Anterior tongue *** (13.3%), FoM (43.3%), lip (6.7%), gum (16.7%), palate (13.3%), cheek/vestibule/retromolar (6.7%)	HPV DNA ISH and p16 IHC (>70%)
The Netherlands, 2021 [41]	Vanderbilt University and Medical Center, and Erasmus Medical Center Rotterdam. The Dutch Cancer Registries	2008–2014	940	59	64	**-**	1:1.4	2.2	Mobile tongue (28.6%) *, FoM (57.1%), vestibulum of mouth (4.8%), cheek mucosa (9.8%), retromolar trigone (0%)	HPV DNA PCR and HPV RNA PCR and p16 IHC (>70%)
Poland, 2020 [39]	Maria Sklodowska-Curie Institute Oncology Center, Cracow	1991–2014	25	-	-	-	-	16	-	HPV DNA PCR and p16 IHC (>75%)
Switzerland, 2019 [42]	The University Hospital of Geneva	2001–2011	152	-	-	-	-	0.66	Mobile tongue (58%) *, lower lip (0%), upper gingiva (8.3%), FoM (25%), buccal mucosa (0%), alveolar ridge (8.3%), retromolar trigone (0%), hard palate (0%) ^d^	HPV DNA PCR and p16 IHC (>70%)
UK, 2021 [36]	Eastman Dental Institute and University College London	1986–2004	116	-	-	-	-	0	0%	HPV DNA PCR
**Middle East**										
Jordan, 2020 [24]	King Hussein Cancer Center and King Hussein Medical Center	2013–2018	27	-	-	-	-	37	Tongue (100%) **, FoM (0%), buccal (0%)	HPV DNA PCR
Saudi Arabia, 2019 [21]	King Faisal Specialist Hospital and Research Centre	2002–2016	257	-	-	59.2	1:1.4	2	Oral tongue (25%) **, FoM (0%), retromolar (50%), buccal (25%), hard palate (0%)	HPV DNA PCR and p16 IHC (>70%)
Turkey, 2021 [26]	The Turkish Ministry of Health	2013–2017	34	-	-	-	-	17.6	Oral tongue (16.7%) **, FoM (0%), buccal mucosa-lip (66.7%), hard palate (16.7%)	HPV DNA PCR
**North America**										
Mexico, 2021 [47]	The Unidad Médica de Alta Especialidad de Oncología del Centro Médico Nacional Siglo XXI, of the Instituto Mexicano del Seguro Social	2011–2017	171	-	-	-	-	9.6	-	HPV DNA PCR and p16 IHC (>70%)
USA, 2017 [46]	The Johns Hopkins Hospital Sydney Kimmel Comprehensive Cancer Center and The University of California– San Francisco Helen Diller Family Comprehensive Cancer Center and affiliated hospitals	1995–2012	253	-	-	-	-	2	-	HPV DNA ISH and HPV RNA ISH
**South America**										
Brazil, 2018 [45]	Santa Rita de Cassia Hospital and University Hospital Antonio Cassiano de Moraes	2012–2015	90	61	57.5	57.9	1:3.1	3.3	Tongue (66.7%) ***, FoM (0%), others (palate, retromolar trigone, gum, buccal mucosa, alveolar ridge (33.3%))	HPV DNA PCR
**South America and Europe**										
Brazil and UK, 2020 [35]	Hospital Santa Rita de Cassia and Hospital Universitário Cassiano Antônio de Moraes (Brazil). University Hospitals Coventry & Warwickshire (UK)	2011–2015	99	-	-	60.5	1:3.5	8	Tongue (75%) *, FoM (12.5%), others: gingiva, hard palate, cheek, vestibule of mouth, and retromolar area (12.5%)	HPV RNA ISH

^a^: included soft palate, ^b^: sublocations only detected with HPV DNA PCR, ^c^: basaloid squamous cell carcinoma, ^d^: sublocations only detected with p16 overexpression. * specified mobile tongue, ** questionable specification, *** unspecified. HPV: human papillomavirus, OCSCC: oral cavity squamous cell carcinoma, FoM: floor of mouth, IHC: immunohistochemistry, ISH: in situ hybridization, PCR: polymerase chain reaction, HC2 HR-HPV: hybrid capture 2 high-risk HPV, UK: United Kingdom, USA: the United States of America.

## Data Availability

Not applicable.

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
