# Peer review of "The Prevalence of HPV in Oral Cavity Squamous Cell Carcinoma"

_viruses, 2023, doi:10.3390/v15020451_

Round 1

Reviewer 1 Report

The authors used several criteria to select the published data for this review by focusing on the anatomic sites which seemed not well reported in most studies. Information from systemic review analysis is valuable to inform the public health authorities about the current statue of this lethal cancer. However, grouping the data comprising 5,007 from 24 countries together could be super challenging as you might dilute some important findings unique to one group/country but not for others. For instance, we all understand HPV+OCSCC is an alarming problem in the developed countries including the USA and Canada especially in white males ( https://pubmed.ncbi.nlm.nih.gov/31358520/).

In the current study, one reference was included from the USA and was further diluted in the large pool. Therefore, the conclusion “Thus, HPV is not a necessary factor nor a high-impact risk factor regarding OCSCC development, and the possibility of site misclassification of mobile tongue with root of 324 tongue cannot be excluded.” is not completely justified given the authors’ acknowledgement of “a significant heterogeneity” among the studies included in the current study. The authors could regroup their data according to geographic or income factors and determine whether OCSCC is correlated to these factors. 

Reviewer 2 Report

The submitted manuscript is an important work for getting actual information on HPV prevalence outside the localisation in oropharynx, and perfectly fits into the aims of the special issue. 

The literature search strategy is appropriate and well documented. 

One question:

If available could you comment if there is a survival difference between HPV-positive and negative oral squamous cell carcinoma patients?

Reviewer 3 Report

The authors performed meta-analysis investigating HPV prevalence in oral cavity SCC (OCSCC). Their results demonstrated that the pooled HPV+ OCSCC prevalence worldwide to be 6% (95% CI; 320 3%-10%). They concluded HPV is not a necessary factor nor a high-impact risk factor regarding OCSCC development. Their analysis is logical and powerful to settle the HPV-OCSCC issue. Some minor issues should be reconsidered.

Line 80, What do asterisks mean?

In table 1, Ref [32] and [42] describe sub-location of oral cavity with patient number. Why aren't they shown in the table?
